# The Role of Emerin in Cancer Progression and Metastasis

**DOI:** 10.3390/ijms222011289

**Published:** 2021-10-19

**Authors:** Alexandra G. Liddane, James M. Holaska

**Affiliations:** 1Department of Pharmaceutical Sciences, University of the Sciences in Philadelphia, Philadelphia, PA 19104, USA; aliddane@mail.usciences.edu; 2Department of Biomedical Sciences, Cooper Medical School of Rowan University, Camden, NJ 08028, USA

**Keywords:** nuclear envelope, cancer, emerin, nuclear lamina, lamins

## Abstract

It is commonly recognized in the field that cancer cells exhibit changes in the size and shape of their nuclei. These features often serve as important biomarkers in the diagnosis and prognosis of cancer patients. Nuclear size can significantly impact cell migration due to its incredibly large size. Nuclear structural changes are predicted to regulate cancer cell migration. Nuclear abnormalities are common across a vast spectrum of cancer types, regardless of tissue source, mutational spectrum, and signaling dependencies. The pervasiveness of nuclear alterations suggests that changes in nuclear structure may be crucially linked to the transformation process. The factors driving these nuclear abnormalities, and the functional consequences, are not completely understood. Nuclear envelope proteins play an important role in regulating nuclear size and structure in cancer. Altered expression of nuclear lamina proteins, including emerin, is found in many cancers and this expression is correlated with better clinical outcomes. A model is emerging whereby emerin, as well as other nuclear lamina proteins, binding to the nucleoskeleton regulates the nuclear structure to impact metastasis. In this model, emerin and lamins play a central role in metastatic transformation, since decreased emerin expression during transformation causes the nuclear structural defects required for increased cell migration, intravasation, and extravasation. Herein, we discuss the cellular functions of nuclear lamina proteins, with a particular focus on emerin, and how these functions impact cancer progression and metastasis.

## 1. Introduction

Generally, the nuclear envelope (NE) outline in normal cells is smooth, free of any malformations. In cancer, morphological abnormalities in the NE are common, appearing as irregular folding, deeper grooves, and cytoplasmic inclusions [1]. Notably, altered NE morphology is a crucial part of a pathologist’s assessment of tumor grade, and correlates with prognosis [2,3]. Some studies have indicated that NE irregularities may be a direct result of oncogene activation, a lack of tumor suppressor function, or genomic instability [1,4,5]. Taken together, these findings suggest that changes in the structure and composition of the NE may be regulated by events occurring early in the transformation process, and could thus be directly linked to tumorigenesis. Four protein groups comprise the NE: Nuclear pore complexes (NPCs), the inner and outer nuclear membrane proteins (INM and ONM, respectively), and the nuclear lamina. 

NPCs are large macromolecular assemblies embedded into the NE to form a channel. NPCs are massive, with molecular masses of approximately 120 MDa, and are built from multiple copies of around 30 different nuclear pore proteins called nucleoporins (NUPs) [6]. The core components of the NPC are embedded in the NE and form a scaffold upon which the inner and outer rings of NUPs are assembled [6,7,8]. NUPS are often characterized by phenylalanine-glycine (FG) repeats, which act as a selective permeability barrier to facilitate active nuclear transport across the NE [6,7,8,9,10,11].

The ONM is contiguous with the endoplasmic reticulum (ER) (Figure 1). Despite the lipid continuity between the NE and ER, both ONM and INM are comprised of diverse groups of proteins that are typically not enriched in the ER. The ONM encompasses a large array of integral membrane proteins that contain a KASH (Klarsicht, ANC-1, Syne Homology) domain in their structure [12]. Nuclear envelope spectrin repeat (nesprin)-1 and -2 are two related members of this group and play an important role in nuclear positioning through their interaction with the actin cytoskeleton [12]. Studies have confirmed the presence of a complex of proteins known as the linker of the nucleoskeleton and cytoskeleton (LINC) complexes [13] (Figure 1). The LINC complex is made of large nesprin isoforms located on the ONM binding to cytoskeletal actin or, via plectin, intermediate filaments [14,15,16]. Nesprins have also been reported to interact with microtubules and the centrosome via molecular motor complexes, such as dynein/dynactin and kinesin [17]. The core of the LINC complex is comprised of nesprin interactions with SUN proteins across the perinuclear space. SUN proteins then create a link with the cytoskeleton and nucleoskeleton by binding to INM proteins such as lamins, inner nuclear membrane-localized nesprin isoforms, NPC proteins, chromatin, and other NE proteins [18,19,20,21]. Forces exerted extracellularly or on the cytoskeleton are transmitted through the cytoskeleton to the nucleus, resulting in altered nuclear morphological changes [22]. 

Underlying the INM are lamins, a family of type-V intermediate filament proteins required for the nuclear structure. There are four lamin isoforms in humans: Lamin A, Lamin B1, Lamin B2, and Lamin C. Lamin A and C are alternatively spliced isoforms encoded by *LMNA/C*. Lamin B1 and B2 are encoded by *LMNB1* and *LMNB2*, respectively. A- and B-type lamins appear to form separate, yet interdependent, filament networks in the nucleus. Lamina networks have been implicated in nuclear structures and functions, as well as genomic organization and gene expression [23,24,25]. Cell-stretching and micropipette aspiration experiments have indicated that lamins encoded by *LMNA* (A-type lamins) have a larger impact on nuclear stiffness than lamins encoded by *LMNB1* and *LMNB2* (B-type lamins). It has become clear that nuclear stiffness is strongly correlated with the expression of A-type lamins, although an increased expression of lamin B1 can also increase nuclear rigidity [26,27,28]. A positive relationship exists between tissue elasticity and lamin levels in the nucleoskeleton. This implies that nuclei in stiffer tissues is stiffer due to a higher lamin content [28]. The INM contains a large number of integral INM proteins called NE transmembrane proteins that, together with the lamins, form the nuclear lamina [29,30], but what role these INM proteins play in regulating the nuclear structure remains to be elucidated.

Several studies have reported on the disruption of nuclear lamina proteins in various cancers, including ovarian cancer [31], basal cell carcinoma [32], colorectal carcinoma [33], hepatocellular carcinoma [34], prostate cancer [35], and thyroid cancer [36]. Furthermore, tumor cells often show an aberrant nuclear structure, such as nuclear size and shape, number and sizes of nucleoli, and chromatin texture. These alterations can be characteristic of a given tumor type and stage, and thus, they are often used in cancer diagnosis [37]. These findings are particularly relevant to invading cancer cells, which must move through tissues containing micron-sized spaces often smaller than the size of the nucleus. As changes in the NE correspond to functional changes in the nucleus, morphological changes in the nucleus are thought be involved in metastatic transformation.

## 2. Emerin Domain and Structure

The human emerin gene (*EMD*) consists of six exons and five introns and is located on the X-chromosome. *EMD* encodes a 254 amino acid protein with a 220 amino acid N-terminal nucleoplasmic domain, a 23 amino acid C-terminal transmembrane domain, and an 11-residue luminal domain. Newly synthesized emerin is inserted into the ER post-translationally and then diffuses through the ER into the NE [38,39]. Emerin enters the nucleus by passive diffusion while membrane-anchored [39,40], and its localization is stabilized by binding A-type lamins.

Emerin is ubiquitously expressed [41,42,43] and has been implicated in the regulation of gene expression, cell signaling, and nuclear and genomic architecture [44,45,46]. Emerin, along with Lap2ß and MAN1, is a founding member of the LEM domain proteins. The LEM domain binds to barrier to autointegration factor (BAF) [47,48]. The emerin LEM domain is at its N-terminus (residues 4–44) [48]. BAF can simultaneously bind emerin and DNA. Outside the LEM and transmembrane domains (residues 223–246), emerin has no known secondary structure [49]. Emerin residues 70–178 bind to lamin A [50] and mediate its NE enrichment [51]. Recent studies have also suggested that BAF may be critically important for emerin NE localization, as emerin and lamin A both fail to associate with assembling nuclear envelope in cells that express a dominant mutant of BAF [52]. Thus, BAF is a key NE localization factor for both emerin and lamin A post-mitotically [52]. Together, these finding indicate that the recruitment and retention of emerin may involve sequential interactions with BAF and lamins [52].

The emerin nucleoplasmic region contains the LEM domain and a large intrinsically disordered region (IDR) [49]. Multiple studies have revealed that fragments of emerin bind to itself, suggesting emerin IDR promotes self-association. Experiments have revealed that a fragment comprising emerin residues 1–222 binds to itself, suggesting C-terminally truncated emerin could form homodimers and/or multimers [50,53]. Experiments have shown that emerin 170–220 is sufficient to bind emerin 1–221, whereas emerin 1–160 does not bind. Further studies have shown that the interaction between the LEM domain and the IDR is necessary in order to oligomerize [54]. Further studies will be needed to fully understand the role of emerin’s self-assembly pathway and how it may be influenced by post-translational modifications and partner binding. It is important to keep in mind that emerin is highly phosphorylated in cells, particularly at positions 4, 8, 10, and 28 at the N-termini of helices α1, α2, and α3 [55]. The large number of potential phosphorylation sites is predicted to impact the stability of the LEM domain and regulate emerin self-assembly [54]. Additionally, BAF and lamin binding have been shown to be regulated by emerin self-assembly [50,56]. Modification and oligomerization have been identified as essential events for the regulation of the LEM domain protein network at the nuclear envelope [57]. 

In addition to binding lamin A and BAF, emerin directly binds to at least 14 other proteins [58]. To map emerin functional domains, clustered Ala-substitution mutations have been generated in recombinant emerin. One set targets residues, that are identical between emerin and LAP2β, which is thought to mediate shared functions [52]. Another set of mutations targets residues that differ between emerin and LAP2β; these are predicted to disrupt emerin-specific functions [59]. Four human mutations (S54F, Q133H, P183H, Δ 95–99) have also been tested [60,61]. These four mutations are unique as they cause Emery-Dreifuss Muscular Dystrophy (EDMD), even though the mutant proteins localize normally and are expressed at normal or near-normal levels [60,61]. Emerin polypeptides bearing the various mutations have been tested for binding to as many as eight different binding partners (Figure 2A). This research resulted in a binding map for emerin based on the locations of mutations that disrupt binding to each partner (Figure 2B). 

## 3. Emerin Functions

### 3.1. Transcription Regulation

Emerin binds a number of transcription regulators, including GCL [59], Btf [62], Lmo7 [63], ß-catenin [64], SIKE [65], and BAF [52], and regulates the expression of their target genes. Emerin has been shown to regulate the expression of many muscles and cardiac genes [44,66,67]. Some important emerin-binding transcription factors and how emerin regulates their function are discussed below. 

#### 3.1.1. Germ Cell-Less (GCL)

GCL is a transcription repressor that binds and inactivates the DP3 subunit of the E2F-DP3 heterodimer [68]. GCL binds to emerin via regulator-binding domain (RBD)-1 and RBD-2 in emerin [59]. Emerin downregulation results in mislocalization of GCL from the NE to the cytoplasm, ultimately leading to an increase in E2F-mediated gene expression [59]. These results show that GCL represses E2F- and DP3-dependent transcription when bound to emerin [46]. This implicates emerin in the control of cell proliferation, since E2F-DP3-dependent genes are required for S-phase entry and repressed by retinoblastoma protein (Rb). Further support can be found in the increased proliferation seen in emerin-null cells [64]. GCL also interacts with the protein GAGE, which is upregulated in many cancers [69], and GCL and GAGE are co-expressed in cancer cells [70]. GCL recruits GAGE proteins to the NE in HeLa cells and human cancer cell lines [70]. This suggests that LEM domain proteins or emerin may influence cancer via GCL binding; however, it is unknown the degree that LEM domain proteins regulate cancer-related pathways. 

#### 3.1.2. Bcl-2 Associated Transcription Factor (Btf)

Btf is predicted to play a crucial role in development. Btf-null mice have polydactyly, immunological problems, and incomplete lung development [71]. Btf binds directly to emerin [59,62], whereby it may help regulate the DNA damage response and apoptosis. Btf binds to anti-apoptotic Bcl proteins (Bcl-2 and Bcl-xL) in the cytoplasm [72]. Upon apoptotic induction, Btf is released from Bcl proteins and accumulates at the NE [62,73]. Btf localizes to sites of DNA damage, where it binds to histone H2AX at the site of damage [74] and forms a complex with protein kinase Cð to activate p53 [75]. Btf also regulates transcription via its association with ribonucleoprotein complexes and as an mRNA splicing factor [62,76,77,78]. Thus, emerin binding to Btf may help to regulate the mRNA splice site choice, potentially through emerin’s interaction with the splicing regulator YT521-B [79]. 

#### 3.1.3. Lim Domain Only Protein 7 (Lmo7)

Lmo7 is a transcription factor that passes between the cell surface and the nucleus, where it binds to emerin to inhibit Lmo7 transcriptional activation [63]. Emerin is required for Lmo7 nuclear localization, with emerin downregulation inhibiting Lmo7 nuclear localization [63,80]. Emerin binding to Lmo7 inhibits transcriptional activation, including emerin, suggesting a negative feedback loop in the regulation of emerin expression [63]. Lmo7 is highly expressed in heart and skeletal muscle [81,82]. Lmo7 binds to the promoters of myogenic differentiation genes and activates their expression [83]. Emerin binding to Lmo7 prevents Lmo7 binding to these myogenic promoters, resulting in their transcriptional repression [83]. After myogenic differentiation, Lmo7 is primarily localized to the cytoplasm, where it interacts with focal adhesion proteins [84]. Thus, in addition to regulating important myogenic differentiation genes during differentiation and regeneration, Lmo7 is predicted to play important roles in muscle cell adaptation to mechanical stress. 

Lmo7 also has well-documented roles in cancer pathology. The P100 Lmo7 splice variant (with a truncated C-terminal region) has been identified in Yoshida hepatoma AH130W1 cells treated with transforming growth factor-ß (TGF-ß) [85]. TGF-ß induces alternative splicing of the Lmo7 gene and promotes migration of cells in in vitro invasion assays [85,86,87]. In addition, the increased expression of Lmo7 has been reported in colorectal, breast, liver, lung, pancreas, stomach, and prostate cancer, suggesting that Lmo7 may play a role in carcinogenesis [88,89,90]. Additionally, the Lmo7 gene is located on chromosome 13q22, which has been implicated in hereditary breast cancer [91,92,93]. Finally, Lmo7-deficient mice develop irregular epithelial lesions, which results in the development of lung adenocarcinoma at an older age, suggesting a role for Lmo7 as a tumor-suppressor gene [94]. Despite these findings, the role of Lmo7 in carcinogenesis has yet to be fully understood. 

#### 3.1.4. Barrier-to-Autointegration Factor (BAF/Banf1)

BAF is a highly conserved protein essential for a variety of cellular functions, including post-mitotic nuclear assembly [95], cell viability [47], and cell cycle progression [96,97]. BAF facilitates NE reformation during mitosis. In anaphase, BAF is targeted to chromosomes near the spindle attachment sites. Upon its recruitment, BAF recruits NE proteins, lamin A and emerin, to these regions during NE reassembly [98,99,100]. BAF is ubiquitously expressed, but appears to have tissue-specific roles. *C. elegans* lacking BAF results in tissue-specific impairment, including gonadal cell migration, vulva formation, muscle maintenance, and germ-line survival and maturation [95]. 

BAF is an integral part of the nuclear lamina and directly binds to all LEM domain proteins. BAF also condenses DNA via looping by binding to histones H3 and H4 and selected linker histones, such as H1.1 [101,102]. BAF overexpression has diverse effects on post-translational histone modifications and influences silencing and activating histone marks [103]. The regulation of gene expression by BAF may result from its regulation of chromatin architecture. 

BAF is present in an emerin-containing regulatory complex purified from HeLa cells [65]. This regulatory complex contains histone deacetylases (HDACs) 1 and 3 [65], suggesting that emerin and BAF may interact to repress chromatin at the NE. BAF also forms higher-order complexes with Lap2ß and DNA [104]. This suggests Lap2ß binding to BAF–DNA complexes may recruit chromatin to the NE. Mutations in BAF that cause Nestor-Guillermo progeria syndrome (NGPS) lead to BAF protein instability, resulting in less BAF protein expression [105,106]. These mutations also result in mislocalization of emerin to the cytoplasm [106]. Emerin binding to BAF is required for proper nuclear reassembly after mitosis [99], suggesting emerin mislocalization or its absence may contribute to the NGPS phenotype. 

Emerin and BAF may be important for regulating DNA repair. Evidence of emerin and BAF association with DNA repair proteins (Cul4a and DDB2) suggests that emerin may be important for the DNA damage response [101]. In *C. elegans*, BAF and LEM domain proteins anchor to the INM [107]. Emerin-null *C. elegans* has been found to be hypersensitive to DNA damage [107]. 

### 3.2. Signaling

Due to the vast number of binding partners, it is no surprise that emerin plays a role in many different signaling pathways. These pathways include Wnt, TGF-ß, Notch, IGF [62], JNK, MAPK [108], NF-κB, integrin signaling [109,110,111,112], VEGF, embryonic stem cell signaling, G2M checkpoint signaling, actin-mediated cytoskeletal signaling, HIPPO pathway, DNA damage, mitotic pathways, Ox40 signaling, and Cdc42 signaling [110].

Many types of cancers show a disruption in these signaling pathways. Oncogenic mutations can cause the affected genes to be overexpressed or produce mutated proteins whose activity is dysregulated [113]. Components of developmental signaling pathways such as Wnt, Hedgehog, Hippo, and Notch can also be affected, as can downstream nuclear targets of signaling pathways (i.e., transcription factors, chromatin remodelers, and cell cycle effectors) [113]. However, it is unclear if disruption of these key signaling pathways is caused by a loss of, or mutations in, emerin and to what extent this could contribute to cancer progression and metastasis. Some of these important pathways are discussed further below. 

#### 3.2.1. Wnt/ß-catenin

Emerin directly binds to ß-catenin, the Wnt-signaling transcription factor, through its adenomatous polyposis coli (APC)-like domain [64]. Emerin’s binding to ß-catenin inhibits its activity by preventing the accumulation in the nucleus. Emerin-null cells show increased expression and accumulation of ß-catenin [64,114], leading to an increase of ß-catenin target gene expression and increased cell proliferation [64]. Interestingly, knockdown of ß-catenin leads to decreased mRNA expression and nuclear accumulation of emerin [114]. This suggests that emerin and ß-catenin regulate one another’s expression, localization, and activity. Wnt/ß-catenin signaling participates in the regulation of tumor immunology. Analysis of metastatic human cutaneous melanoma samples has shown that tumor-intrinsic ß-catenin activation excludes T cell infiltration into the melanoma tumor microenvironment [115]. ß-catenin activation may represent one mechanism of resistance to T cell-based immune-oncology therapies. Using the mouse mammary tumor virus (MMTV)-induced breast cancer model, scientists have found that Wnt signaling induces the expansion of stem-like cells during mammary tumor progression [116,117]. Since Wnt/ß-catenin signaling promotes differentiation of many cancer stem cells, it is of a great significance to better understand the underlying mechanisms to reduce stem cell behavior.

#### 3.2.2. Mitogen-Activated Protein Kinase (MAPK)

MAPK signaling kinases phosphorylate cytoplasmic and nuclear targets to regulate diverse cellular processes related to malignant transformation [118]. Dysregulation of MAPK signaling is a common feature in cancer, often occurring downstream of growth factor signaling or through mutations [118]. Interestingly, the expression of emerin mutants, as well as the loss of emerin or lamin A expression, results in hyperactivation of multiple MAPK signaling branches, namely, extracellular signaling-regulated kinase 1/2 (ERK1/2), c-Jun N-terminal kinase (JNK), and p38 MAPK [108,109,110,111,118,119]. The molecular mechanism in which lamin A/C and emerin modulate MAPK signaling remains incompletely understood. A-type lamins interact with ERK1/2 and c-fos. This results in ERK1/2-dependent release of the c-fos transcription factor from lamin A/C and subsequent activation of AP1-mediated transcription and proliferation [111]. Competitive ERK1/2 binding to lamin A also regulates cell cycle progression through movement and inactivation of Rb. Translating these findings into signaling in the context of cancer cells could be important in determining the role of emerin alterations in tumor progression. 

#### 3.2.3. Megakaryoblastic Leukemia Protein-1 (MKL1)/Serum Response Factor (SRF)

MKL1 is a transcription coactivator of SRF that regulates the expression of genes involved in cell migration, growth, and differentiation [120]. MKL1/SRF signaling has key functions in tumor progression, such as mediating TGF-β-induced epithelial–mesenchymal transition (EMT) and promoting cell migration and metastasis. MKL1 is localized in the cytoplasm through binding to G-actin. Mitogenic or mechanical simulation causes actin polymerization, resulting in translocation of MKL1 to the nucleus [121,122]. Increased MKL1 nuclear import, coupled with decreased nuclear export, causes the accumulation of MKL1 in the nucleus, where it co-activates SRF to turn on genes regulating cellular motility and contractility, including vinculin, actin, and SRF itself [123]. MKL1 is also regulated by nuclear actin dynamics. Loss of lamin A/C or emerin from the NE impairs nuclear translocation and signaling of MKL1 due to the role of emerin in controlling nuclear actin polymerization [124,125]. Emerin is a crucial modulator of actin polymerization and loss of emerin from the nuclear envelope disrupts actin dynamics and impaired MKL1 signaling [120,126].

### 3.3. Nuclear Structure

Biophysical studies have provided evidence supporting emerin performing important roles in maintaining nuclear architecture, including emerin-null cells exhibiting decreased elastic and more malleable NEs [127,128]. At the cellular level, cultured lamin A/C- or emerin-null mouse embryonic fibroblasts (MEFs) also show nuclear morphology defects, including increased nuclear deformability, impaired viability under mechanical strain, and defective mechanotransduction [128,129]. 

#### 3.3.1. Lamins

A comprehensive review of the role of lamins in maintaining nuclear structure is beyond the scope of this review. Briefly, there is a nuclear intermediate filament network composed of lamins A, B, and C that forms a nuclear envelope-associated lattice and provides the nuclear envelope its strength [41,52,130,131]. The lamin network is required for stable localization and retention of inner nuclear membrane proteins, including emerin [41,52]. There is overwhelming evidence supporting the role of the nuclear lamina in the mechanical support of the nucleus. Additionally, changes in nuclear lamina composition have been implicated in a variety of diseases and has major effects on the mechanical response of the cells as a whole. MEFs lacking lamin A and C show defective shapes and decreased stiffness of nuclei [26,129]. Meanwhile, MEFs lacking only lamin A have only slight alterations in nuclear shape and stiffness, suggesting complementary roles in the mechanical support of the nucleus [129]. In contract, MEFs lacking lamin B1 show alterations in nuclear shape, but no change in nuclear stiffness [129]. 

Alterations in nuclear lamina composition can also result in mechanically induced changes in gene expression. In response to mechanical stress, MEFs lacking lamins A/C exhibit abnormal signaling leading to attenuated NF-kβ-regulated transcription and impaired activation of mechanoresponsive genes [129]. Cells lacking emerin also result in impaired cellular signaling in response to mechanical stimulation, where the expression of NF-kβ-regulated genes is impaired in response to mechanical stimulation [127,128].

#### 3.3.2. The LINC Complex, Nesprins, and SUN Domain Proteins

The LINC complex physically connects the cytoskeleton to the nucleus and transmits mechanical forces from the cytoskeleton across the NE to the nucleoskeleton (Figure 3) [132,133]. The LINC complex has been implicated in cell division [134], cytoskeletal organization [135], and organelle positioning [136], showing its importance for basic cellular functions. The ONM components of the LINC complex are the nesprins, a large family of spectrin-repeat transmembrane proteins [135,137], which bind to the cytoskeleton via actin or microtubules [135,138]. Nesprins interact with the C-terminus of INM SUN domain proteins in the lumen of the NE [139,140]. SUN domain proteins are essential for the recruitment of nesprins to the ONM [141,142]. The interactions between SUN domain proteins and nesprins maintain the size of the lumen [13,30] and aid in the positioning of the nucleus in mature myofibers [143].

The characteristic feature of the SUN domain family proteins is a 50 amino acid domain that is conserved between *S. pombe* (Sad1 protein) [144] and *C. elegans* (UNC84) [145]. Mammalian cells have five SUN domain proteins, with SUN1 and SUN2 present on the NE in somatic cells [146]. SUN1 and SUN2 proteins consist of the helical N-terminal domain that can bind to lamins [140] and nuclear pore complex proteins [147,148], a single pass transmembrane domain that anchors the protein in the INM [149], a luminal helical domain required for trimerization of SUN proteins [15], and the C-terminal SUN domain, which interacts with the KASH domain of nesprins [13]. 

Mammals have four nesprins (genes *SYNE 1–4*), with nesprins 1–3 having multiple isoforms resulting from alternative splicing, initiation, and termination [22,150,151]. The expression of various nesprin isoforms is highly tissue-specific [151]. All nesprins contain a central region of multiple spectrin domains, with the number of repeats varying between isoforms [135]. The C-terminus of all nesprins, but not all isoforms, contains a ~60 amino acid KASH domain. This domain, consisting of a transmembrane domain and a short, luminal domain, is essential for anchoring nesprins to the NE [23,135]. Typically, the N-terminal domain of nesprins contains specific motifs to interact with different cytoskeletal proteins. For example, the N-terminal domain of the nesprin-1 and -2 “giant” isoforms (1000 and 800 kDa in size, respectively) contains an actin-binding domain [135,152,153]. Nesprins-1 and -2 can also interact with microtubule-associated motors, dynein/dynactin, and kinesin [23]. Nesprin-3 can connect to intermediate filaments via plectin [154], while nesprin-4 binds the microtubule-associated motor kinesin [155]. While localization of larger nesprin isoforms is restricted to the ONM, shorter isoforms can also be present at the INM, where they can interact with lamins and emerin [156,157,158]. 

The application of force to the plasma membrane or cytoskeleton deforms the nucleus and activates mechanosensitive genes [159,160]. LINC complex proteins and lamin A/C directly transmit mechanical force from the plasma membrane and cytoplasm to the nucleus [129,133]. Emerin and lamin A directly bind nesprins and SUN domain proteins (Figure 3) [13,19,157]. Emerin-null or lamin A/C-null cells have defects in mechanotransduction and show increased nuclear fragility [127,129]. Emerin is also required for the activation of the downstream mechanosensitive genes IEX-1 and EGR-1 [128]. More recently, disruption of the LINC complex was shown to impair extracellular mechanical cues of chromatin stretch and transcription [161] and nuclear translocation of transcription cofactors [162,163,164]. Genetically encoded biosensors of tension in nesprins now exist [165], and direct force application on nesprins has been shown to elicit nucleus-autonomous signaling that targets nucleus stiffness [166]. An additional mechanism by which lamins and emerin can affect mechanotransduction signaling has been identified. The actin polymerization-promoting activity of emerin at the NE can influence nuclear and cytoskeletal actin dynamics to modulate localization and activity of the mechanosensitive transcription factor, MKL1 [120]. Yet it is still unclear what role emerin plays in the mechanotransduction process and how the disruption of emerin changes the structural integrity of the nucleus.

#### 3.3.3. Actin

Studies to better understand the structure and composition of the NE have led to the discovery that the nuclear interior contains actin [167,168], myosin I [169,170], and αII-spectrin [171]. Actin oligomers or short polymers can be found in the nucleus [172,173,174]. All isoforms of actin contain nuclear export sequences [175], which are thought to prevent the spontaneous assembly of actin filaments inside the nucleus. The structural organization of nuclear actin remains unclear [176]. Nevertheless, nuclear actin has been implicated in a number of functions highly relevant to tumorigenesis [177].

Nuclear actin has numerous known functions, including DNA organization, orientation, and stabilization, responding to cellular stress, organization of gene regulator complexes, transcription, RNA synthesis, and nuclear export [168,178]. Actin has also been shown to regulate nuclear architecture through interactions with nuclear pore-linked filaments and NE proteins [179]. Emerin binds to actin at the pointed end of actin filaments to stabilize F-actin in vitro [125]. In addition to actin, emerin also binds other nuclear structural components, including nuclear myosin I and the nuclear-specific spectrin isoform α-II [65]. Emerin has also been found to co-purify with nuclear protein 4.1R, which is known to bind spectrin and actin [180]. 4.1R is also required for mitotic spindle formation and nuclear assembly [181,182]. Reduced emerin expression results in less 4.1R at the INM [64,183]. Thus, emerin is predicted to play a role in the formation of cortical nuclear actin–myosin networks near the NE, which is thought to provide structural rigidity to the NE by forming a strut-like complex (Figure 3).

### 3.4. Chromatin Architecture 

Many studies have shown that emerin-null cells have less repressed chromatin [184,185,186]. Additionally, fibroblasts [184] and skeletal muscle [187] from EDMD patients have altered genomic organization. Chromatin modifications indicative of relaxed chromatin are increased in emerin-null myogenic progenitors [188]. Cells exposed to softer extracellular matrices will reposition chromosome territories in an emerin-dependent manner [189]. Together, these data support a role in which emerin regulates chromatin repression; however, the mechanism remains unclear. Repressive chromatin is established, anchored, and maintained at the NE via the nuclear lamina [190,191,192,193,194,195,196,197]. Simply moving actively transcribed genes to the nuclear periphery results in their repression [191,195]. BAF and the nuclear lamina localize to specific regions of the chromatin at the end of mitosis during NE reformation [99]. This suggests that their recruitment to chromatin may recruit repressed loci to the NE during reassembly. 

Global mapping of chromatin interactions with B-type lamins or emerin has shown that approximately 40% of the *Drosophila* and human genomes contact the NE in large (0.1–10 Mb [192]) discrete regions named lamin-associated domains (LADs) [192,198]. LADs are enriched in heterochromatin [192,193] and are characterized by repetitive DNA, low gene density, and repressive chromatin marks [199,200]. LAD organization is highly dynamic and can be altered in response to extracellular signaling and cell differentiation [200,201].

LADs can be defined as constitutive LADs (cLADs) and facultative LADs (fLADs). cLADs are always associated with the nuclear lamina, while fLADs vary by cell type [202]. Interestingly, LADs have broad overlap in genomic coverage in different cell types [200]. During the differentiation of murine embryonic stem cells (ESCs) into neural precursor cells and astrocytes (ACs), LADs containing genes that are activated during differentiation lose association with the nuclear lamina, while others are recruited to the nuclear lamina to become repressed [200]. This suggests that LAD association is important for differentiation and cellular identity.

Some of the interactions between the genome and the NE are sequence-dependent, since cLADs are high in A/T content [202]. Research has found these lamina-associated sequences (LASs) are co-localized with lamin B during NE assembly and are sufficient to repress lamin-associated genes [203]. One such LAS, an extended GAGA motif, is necessary to localize these genomic loci to the nuclear lamina to silence. This is facilitated via binding of cKrox to the GAGA sequence, which then associates with Lap2ß through HDAC3, resulting in association and anchoring to the nuclear lamina [203]. To what extent the repression of chromatin at the NE is sequence-dependent remains an open question. 

How LADs are established or maintained in the NE is unclear, but more clues are beginning to emerge. Emerin associates with the nuclear co-repressor (NCoR) complex, which represses genes by stably binding chromatin [65]. The catalytic component of the NCoR complex is HDAC3, which deacetylates specific lysine (e.g., K5ac) residues in the histone H4 (H4K5ac) tail to promote NCoR interaction with chromatin [188]. Emerin also binds HDAC3 directly. Furthermore, emerin association increases the enzymatic activity of HDAC3 by 2.5-fold in vitro, suggesting that emerin enhances HDAC3-dependent gene silencing [188]. This finding is consistent with the epigenetic phenotype (globally increased H4K5 acetylation) seen in emerin-downregulated cells and emerin-null mouse fibroblasts [188]. Thus, the HDAC3–emerin association may be fundamentally important for tissue-specific gene repression. 

### 3.5. Summary of Emerin Function

Emerin is involved in a diverse range of biological processes. These includes transcription regulation, cell signaling, nucleo-cytoskeletal mechanotransduction, nuclear structure, chromatin compaction, genomic organization, and epigenetic modification. Due to emerin’s vast number of binding partners, there are numerous functions of emerin across many different cell types (Table 1) [65].

## 4. Relevance of Nuclear Mechanics and Mechanotransduction in Cancer Progression

With growing advances in the understanding of the physics of cell motility, the mechanical properties of cancer cells have become an increasing area of interest [207]. The properties of the nucleus can dominate the overall cellular mechanical response when cells are subjected to large deformations, as the nucleus is the largest and stiffest organelle, often occupying a large fraction of the cell’s volume [207]. Several lines of evidence, suggest the ability of the nucleus to deform, impose a rate-limiting step in non-proteolytic cell migration in 3D environments [208,209]. In this section, we summarize changes in the nuclear structure and morphology observed in various cancers and describe the role of nuclear deformability in cell migration. In addition, we discuss the intricate feedback between the mechanics of the cellular microenvironment and intracellular organization and function.

### 4.1. Altered Nuclear Structure and Morphology in Cancer Cells

The shape, size, protein composition, and texture of the nucleus are often altered in malignant cells. The nucleus may acquire grooves, folds, or indentations; the chromatin may aggregate or disperse; and the nucleolus may become enlarged [24,37]. In normal cells, the nucleus is often round with smooth outlines, but in cancer cells, the outline is often irregular. Different combinations of nuclear abnormalities are characteristic of different cancer types, and nuclear appearances are often used for cancer diagnosis and staging. Similar alterations in nuclear morphology are seen in cells lacking specific NE proteins or expressing mutant NE proteins, suggesting a possible link between dysregulated NE proteins and cancer pathology [37,210,211]. 

Lamins are commonly reported to show altered expression in human tumors, especially those with malignant phenotypes. Interestingly, the metastatic potential of different cancers often correlates with specific expression profiles of nuclear lamina proteins [32,212,213,214,215]. For example, reduced lamin A/C expression is correlated with poor prognosis for patients with gastric carcinoma [216]. This is also seen in patients with stage II and III colon cancer who have a significantly increased risk of cancer recurrence [217]. In contrast, another study found patients with increased expression of lamin A/C in colorectal cancer tumors are twice as likely to die compared to patients with tumors negative for lamin A/C [33,218]. Lamin A/C expression levels are highly heterogeneous, even within single tumors or cancer cell lines [21,219], further complicating matters. 

In addition to lamins, other NE proteins have been implicated in a variety of cancers. Genetic alterations of the genes encoding nesprin-1, nesprin-2, and lamin A have been found in several patients with either breast, colorectal, or ovarian cancer [220,221]. Other studies have reported an association between the downregulation or mutations of nesprins and increased risk of invasive ovarian cancer [222]. Expression of several nucleoporin proteins (NUP88 and NUP98) also correlate with aggressive tumor phenotypes [223] and could be used as prognostic markers of disease [224]. Lastly, emerin has emerged as a likely mediator of nuclear shape stability in prostate, lung, and breast cancer [114,225].

Questions remain, such as how does altering NE composition affect nuclear mechanics? The main components controlling nuclear deformability is lamin expression and chromatin organization. Changes in nuclear architecture results in altered shape, rigidity, and mechanics. The nuclear abnormalities seen in cancer cells may allow metastatic cells to pass through narrow constrictions within tissues and in the vascular endothelium more readily. Disruptions in the nucleo-cytoskeletal coupling that could occur by mutations in nuclear lamina proteins has been found to impair mechanical signaling needed during migration [209]. This suggests a balancing act between nuclear compressibility and signaling needs to exist for proper cell function. Though softer nuclei migrate more readily, these nuclei are also more fragile and prone to rupture. Repeated nuclear rupture leads to decreased DNA stability and increased DNA damage. Therefore, changes in nuclear organization could have extensive consequences on gene expression and DNA stability with important implications in cancer progression [226,227,228]. 

### 4.2. The Nuclear Lamina and the Mechanical Tumor Microenvironment

It is becoming apparent that the mechanical tissue microenvironment plays a crucial role in tumorigenesis and tumor progression [229,230]. The tumor microenvironment is composed of various cell types, extracellular matrix (ECM) proteins, blood vessels, lymphatic vessels, and soluble factors, which together create a niche that can support or hinder tumor progression [231]. During tumorigenesis, remodeling the cellular and ECM architecture in the tissue alters the function of tumor and stromal cells, which can further remodel the microenvironment [232]. In particular, ECM composition and rigidity modulate signaling pathways associated with tumor progression, such as ERK TGFβ, and PI3K, thereby affecting EMT and metastasis [229,230]. Increased stiffness in the stroma surrounding the tumor has been shown to accompany tumor progression in a mouse mammary tumor model, and increasing stiffness in vitro is sufficient to convert mammary epithelial cells to an invasive malignant phenotype [233,234]. 

Recent studies have suggested that lamins could play important roles in this mechanosensitive process [25]. Since lamins play a central role in the mechano-regulation of gene expression, changes in lamin levels could influence how cells interpret and respond to changes in their mechanical environment [120,128,129,166]. A proteomic analysis of soft and stiff tissues revealed that A-type lamins increase with tissue stiffness, whereas B-type lamins exhibit a fairly constant abundance; increased lamin A/C likely contributes to lineage specification during differentiation [28]. Furthermore, xenograft tumors of U251 glioblastoma cells exhibit higher lamin A/C levels when grown in the stiffer subcutaneous flank compared to the brain, suggesting that A-type lamin levels can adjust to tissue stiffness in vivo [28]. In mesenchymal stem cells, as matrix rigidity increases, so does A-type lamin expression [235]. However, it cannot be excluded that lamin A/C levels could be affected by other differences in the properties of these microenvironments, such as levels of tissue-specific growth factors, local metabolite concentrations, or other signaling pathways. The physical properties of the microenvironment can also affect nuclear lamina organization. For example, as substrate stiffness increases, cell spreading and lamin A/C epitope in the Ig domain becomes masked [236], suggesting that structural reorganization of the nuclear lamina could further impact interactions with chromatin and other binding partners. These studies demonstrate that both the structural organization and the levels of A-type lamins are dynamically regulated and can modulate cellular mechanotransduction signaling in response to changes in the physical microenvironment.

The composition of the microenvironment may also be affected by changes in nuclear lamina composition. For example, loss of lamin A/C increases collagen production in MEFs [237,238,239,240]. This suggests that cancer-associated changes in the nuclear lamina could remodel the tissue structure and microenvironment to induce changes in surrounding cells and tissues. Further research is required to understand the interplay between nuclear lamina configuration and ECM alterations in relation to cancer cell mechanics and tumor pathology.

Lamin A/C levels would be expected to vary between tumor microenvironments of varying rigidities and thus effect metastatic transformation. Indeed, increased tissue rigidity has been shown to vary between subtypes and within single tumors [241]. In breast cancers, increased rigidity is frequently observed at the tumor edge, with greater variability at the invasive front [241,242]. Increased stiffness is also known to promote EMT-like phenotypes [229]. These findings support the hypothesis that tissue rigidity promotes invasion and metastasis [229,230]. However, invasive breast tumors have more heterogeneity in tissue stiffness than benign tumors, which have a more uniform increase in stiffness over normal tissues [242]. Interestingly, there seems to be a connection between metastatic potential and tumor microenvironment rigidity. Mouse mammary tumors formed in a more compliant tumor microenvironment display increased metastasis, and lung metastases display lower stiffness than matched primary tumors [242]. 

It is intriguing to speculate how the role of lamin A/C in differentiation and plasticity could be misregulated in these different tumor microenvironments. Increased tissue stiffness during tumorigenesis could alter lamin A/C levels, disrupting a variety of functions, including chromatin organization and gene expression. This is particularly important during metastasis, where the switch from an epithelial to a mesenchymal phenotype is thought to be the rate-limiting step in the metastatic cascade [243,244]. This implicates phenotypic flexibility through the modulation of lamin levels and nuclear lamina organization as a driver of tumor progression. Increased microenvironment rigidity may promote invasion from the primary tumor, while softer microenvironments at metastatic sites lead to alterations of and decreases in lamin A/C levels to support the cell-associated plasticity required for growth and metastasis [118]. 

### 4.3. Nuclear Deformability and Cell Migration

Cancer cells often exhibit abnormally shaped nuclei. Curiously, these nuclei resemble those from cells lacking lamin or emerin expression, as well as cells containing certain lamin or emerin mutants. Furthermore, metastatic cancer cell nuclei resemble lamin- and emerin-deficient nuclei, as metastatic cell nuclei are 70% softer than benign cancer cell nuclei [245,246,247]. Decreased lamin A/C expression during carcinogenesis is thought to contribute to these changes in nuclear malleability [209]. These nuclear changes are important, because metastatic cancer cells must undergo large elastic deformations to intravasate into and extravasate out of the vasculature through 1–5 µm slits in the endothelium (Figure 4) [248,249,250]. The nucleus (10–20 µm in diameter) is the largest and stiffest organelle in the cell and thus represents the major bottleneck in this process. The cytoskeleton, on the other hand, is quite flexible, and thus cytoskeletal protrusions can invade spaces as small as 1 µm in diameter [251,252]. Nuclear deformations have been reported in vivo during cancer cell migration and invasion, and have been shown to be the rate-limiting step during proteolysis-independent cell migration [208,209]. A similar size-dependent effect has been observed when studying cell migration in microchannels, as migration speeds progressively decrease below a channel width of 20 µm, with a 70% reduction in migration speed in 3 µm-wide channels [253]. Interestingly, the shapes of these constrictions are not important; rather migration is dependent on its cross-sectional area, which has to be greater than 10% of the cross-sectional area of the non-deformed nucleus [208]. Other studies have observed a reduction in nuclear volume by up to 60% during migration [254] or by 20–40% in micropipette aspiration experiments [127,255], suggesting that maximal nuclear compressibility is a major contributor to confined migration. Maximal compressibility can be defined as the size in which the solid fraction of the nucleus and its contents can be compressed once all empty space has been removed.

These studies illustrate that nuclear malleability plays a key role during confined cell migration, which is controlled, at least in part, by nuclear lamina proteins. The role of neutrophils in immune surveillance requires them to intravasate and extravasate, and thus, they possess highly malleable nuclei. Interestingly, neutrophils have low levels of lamin A/C, as well as other INM proteins [209]. Lamin A overexpression causes stiffer nuclei and impairs confined migration [257]. The expression of progerin, a lamin A mutant that increases nuclear stiffness and rigidity, in fibroblasts also impairs confined migration [254,258,259]. Furthermore, reduced emerin expression alters nuclear mechanics [127,128] and is associated with increased migration and increased metastasis [260]. The expression of emerin impairs confined migration and metastasis, whereas emerin mutants that fail to bind the nucleoskeleton fail to rescue migration and metastasis [260]. The induction of EMT has also been found to reduce the protein expression of some nuclear lamina components, including emerin and lamins, resulting in compromised nuclear envelope integrity [261]. Thus, the linkage of the nuclear lamina and nucleoskeleton to the NE via emerin is important for the nuclear structure to impact confined cell migration and metastasis [262].

### 4.4. Nuclear Rupture of Cancer Cells

As described earlier, the NE forms a well-defined barrier between the nucleoplasm and the cytoplasm and acts as a protective shield for the genetic material. In normal cells, NE breakdown and reassembly is limited to mitosis and is precisely regulated [263]. In many cancer cells, the NE transiently ruptures and then reseals, resulting in a temporary exchange between the nucleus and cytoplasm [264]. Micronuclei-like structures can form as a result of NE rupture, as a portion of chromatin and nucleoplasmic proteins exit the nuclear interior to form a separate and smaller nucleus [264]. Small defects in the nuclear lamina cause the frequency of nuclear rupture events to increase [264]. Other studies have reported increased nuclear fragility and spontaneous and transient nuclear rupture in lamin A/C-deficient mouse embryonic fibroblasts [129,265]. Transient nuclear rupture has been frequently observed in cancer cells while migrating through narrow (2 × 5 µm) microfluidic constrictions. Similar rates of nuclear rupture are seen in lamin-deficient cells [24,266]. Repetitive nuclear rupture results in increased genomic instability and chromatin rearrangements, which could further contribute to cancer progression. 

### 4.5. Changes in Chromatin Organization in Cancer Cells

The genome and associated proteins behave like a spring, elastically resisting micron-sized deformations in the mechanical response of the nucleus [267,268,269,270,271]. Chromatin is an irregularly compacted fluid filling the nucleus that interacts with itself and the nuclear periphery [192]. Variations in compaction may correspond to variations in chromatin stiffness [271,272] and viscoelasticity [273,274,275,276]. A variety of mechanical measurements have demonstrated chromatin as a stiff mechanical element [267,268,270,271,277,278]. Chromatin perturbations, such as blebbing and ruptures, have been observed independent of lamin alterations (Figure 5). Histone modifications are the main driver of chromatin alterations. Changes in heterochromatin and euchromatin compaction levels are commonly seen in cancer and other diseases [279]. Histone modifications that generally increase euchromatin and decrease heterochromatin have been seen to weaken nuclear rigidity, causing abnormal nuclear morphology and nuclear ruptures [280]. In contrast, increasing histone modifications indicative of heterochromatin can rescue the nuclear shape and rigidity. Increased heterochromatin has been found to rescue the abnormal nuclear morphology caused by chromatin and lamin perturbations, including cells with lamin B1 depletion or mutant lamin A (progerin) overexpression [279,280]. Thus, the nuclear morphology caused by different molecular mechanisms can be restored with changes in histone modification and chromatin compaction independent of lamins (Figure 5).

Epigenetic changes in chromatin regulation and compaction can directly impact nuclear stiffness. Thus, the chromatin state alterations frequently observed in cancer cells, including disturbed heterochromatin organization [37], are thought to be associated with altered nuclear deformability. As previously described, there is a dynamic interplay between NE proteins and chromatin organization. Lamin A regulates the dynamics of heterochromatin proteins in early embryonic stem cells [281]; lamin A/C deficiency and mutations in *LMNA* results in a loss of heterochromatin [130,282]. Lamins and emerin play an important role in tethering specific chromatin regions to the nuclear periphery [283,284], which typically serves as a transcriptionally repressive environment [227]. Emerin directly associates with chromatin modifiers and transcriptional repressors, such as Btf [62], HDAC3 [188], and the transcriptional repressor GCL [59]. Furthermore, the interaction of emerin with HDAC3 is important for invasive breast cancer cell invasion and metastasis [260]. There are two possible explanations for these findings: (1) Altered expression of NE proteins directly affect chromatin organization and gene expression or (2) changes in NE composition are the consequence of altered chromatin organization. Both explanations result in altered nuclear mechanics, but understanding the root cause would allow for a more well-rounded understanding of the metastatic transformation of cancer cells. 

## 5. Conclusions

It has long been recognized that cancer cells exhibit changes in the size and shape of their nuclei, and these features serve as important biomarkers in the diagnosis and prognosis of cancer patients. It was recognized as early as 1943, when George Papanicolaou published his book *Diagnosis of Uterine Cancer by Vaginal Smear*, which laid the basis for the now abundant “pap smear” to detect early signs of cervical cancer. The broad prevalence of these changes is particularly intriguing, since nuclear abnormalities are common across a wide spectrum of cancer types, regardless of tissue source, mutational spectrum, and signaling dependencies. The frequency of nuclear alterations would thus suggest that changes in nuclear structure may be crucially linked to the transformation process. However, the factors driving these nuclear abnormalities, and the associated functional consequences, are not completely understood. 

Emerging evidence shows that the mechanical properties of the cell nucleus, particularly its deformability and connection to the cytoskeleton, may play an important role in cancer metastasis. The idea that deformation of the large and stiff nucleus presents a rate-limiting factor during the passage of metastatic cancer cells through narrow constrictions has recently found increasing experimental support [208]. Research continues to that show altered expression and mutations in NE proteins are responsible for determining nuclear stiffness. Additionally, NE proteins are often misexpressed or mutated in cancer.

Emerin’s diverse set of functions in transcription regulation, cell signaling, genomic organization, nuclear architecture, and mechanotransduction leads to an important and interesting question. Through which functions does emerin play a role in metastatic transformation? It is likely that varying combinations of altered cellular mechanics, cell signaling, and mechanotransduction contribute to the increasingly emerging role of emerin in cancer progression. One of the inherent difficulties in understanding the role emerin that plays in metastatic transformation is the diverse array of cellular functions emerin influences and the large number of partners it may act through. 

The NE is tightly regulated, and determining the function of emerin in cancer will require studies that can distinguish between the causes and consequences of altered emerin in particular tumor types and disease stages. It will be important to determine whether emerin expression is different between metastatic subpopulations compared to the primary tumor. If these expected differences are present, it would suggest that emerin could be useful as a biomarker to identify those cells likely to metastasize. Furthermore, determining emerin function throughout the full scope of disease will lead to important insights into metastatic transformation. It is still unclear whether the emerin levels in cancer are a dynamic representation of conditions in the tumor at a particular time, and thus a reflection of the response to transformed signaling pathways and microenvironmental conditions, or whether altered emerin levels in cancer are uncoupled from the normal regulatory mechanism and can independently initiate oncogenic changes. These are important knowledge gaps that need to be addressed.

## Figures and Tables

**Figure 1 ijms-22-11289-f001:**
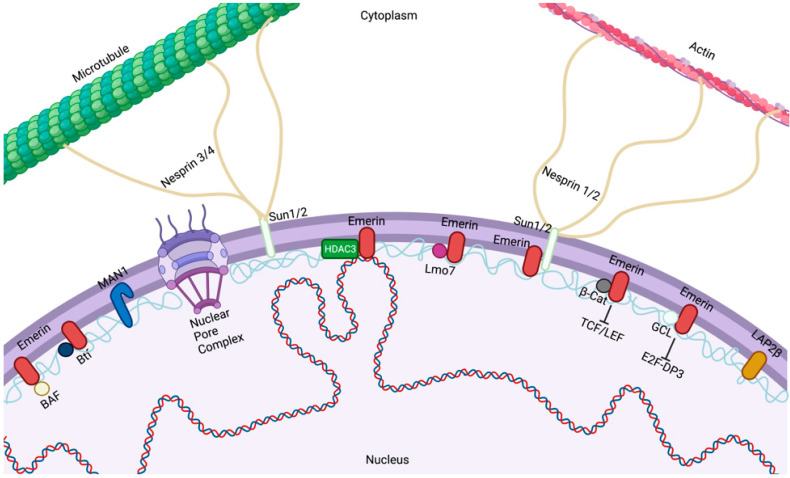
The nuclear envelope. Emerin, Lap2β, and MAN1 bind to lamins at the inner nuclear membrane of the nuclear envelope, where they perform diverse functions within the nucleus. Emerin regulates gene expression by regulating chromatin architecture through binding to HDAC3 and transcription factors (GCL, β-catenin, Lmo7, and Btf) and altering signaling pathways. Created with BioRender.com.

**Figure 2 ijms-22-11289-f002:**
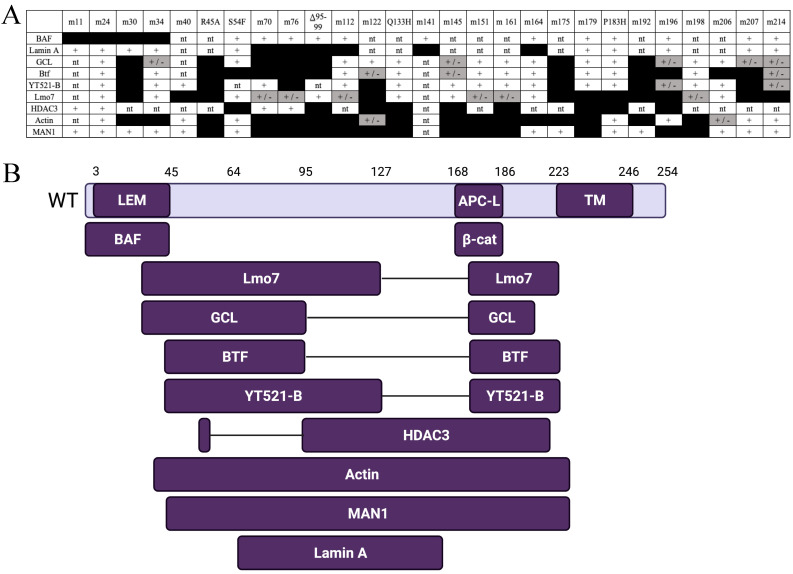
Functional map based on emerin missense mutations that disrupt binding to specific partners. (**A**) Summary of binding results for each named partner, tested for binding to each mutation. Scoring: Normal binding (+), weakened binding (± and gray), and undetectable binding (black box). nt, not tested. (**B**) Results from (**A**) mapped to the emerin polypeptide. APC-L, APC-like domain; TM transmembrane domain; LEM, Lap2, emerin, MAN1, domain [58]. Created with BioRender.com.

**Figure 3 ijms-22-11289-f003:**
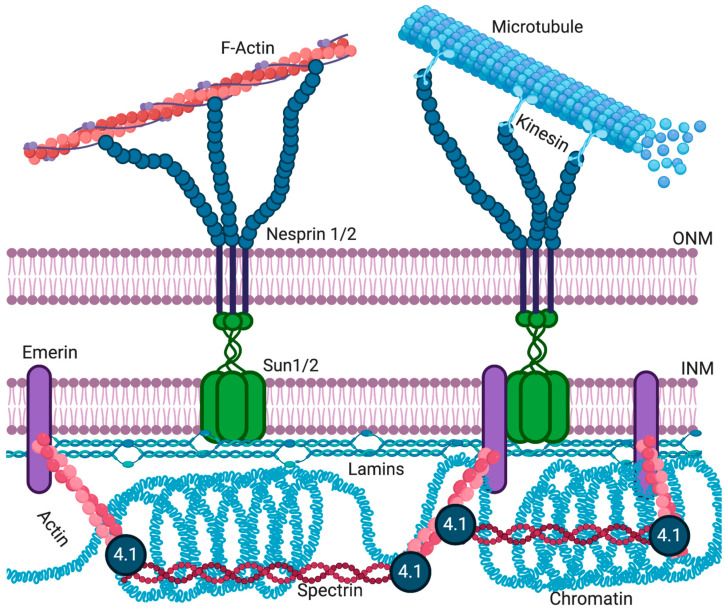
The LINC complex. SUN domain trimers interact with emerin and lamins at the INM. SUN domain proteins interact with KASH domain proteins (nesprins) in the periplasmic space. KASH domain proteins span into the cytoplasm to interact with the cytoskeleton, thereby connecting the nucleus to the cytoskeleton [125]. Created with BioRender.com.

**Figure 4 ijms-22-11289-f004:**
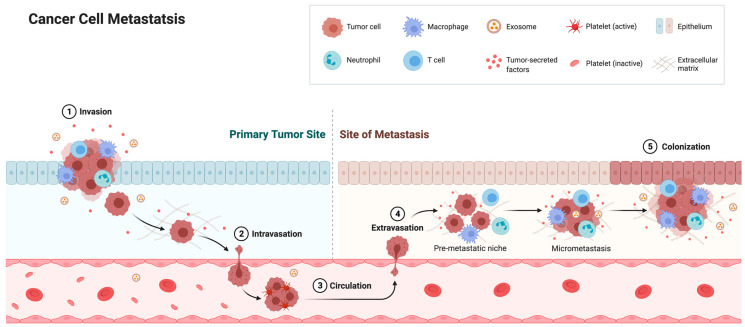
Cancer cell migration and metastasis. Cancer cells grow and invade the surrounding vasculature. Intravasation requires cells to undergo extensive deformations to squeeze through pores between 1 and 5 μm in diameter. Cells travel to a distant metastatic site, where they extravasate and proliferate [256]. Created with BioRender.com.

**Figure 5 ijms-22-11289-f005:**
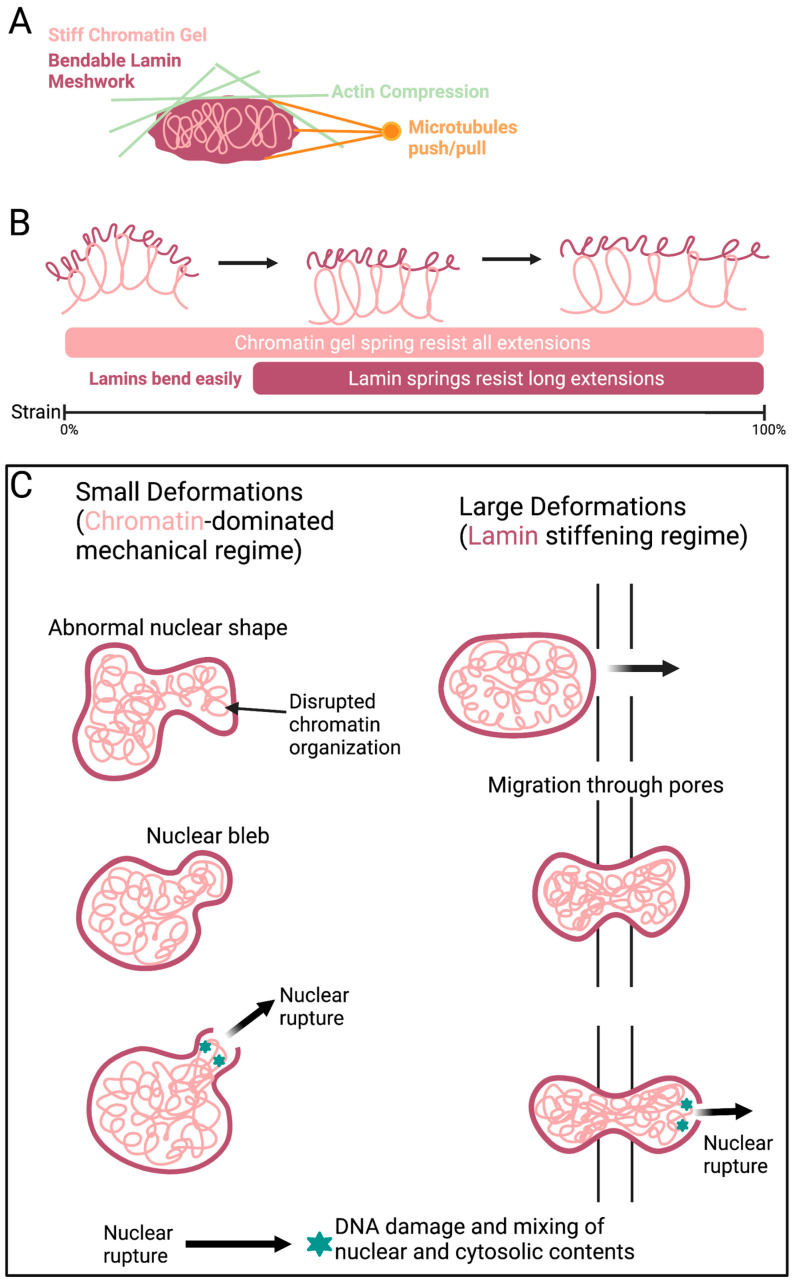
Chromatin is a major contributor to nuclear mechanics. (**A**) Chromatin (pink) and lamins (red) play key roles in the mechanical protection of the nuclear shape and stability. Actin (green) and microtubules (orange) antagonize nuclear shape stability. (**B**) A schematic showing the force response of the nucleus during large and small strains. For small strains, the chromatin acts as a spring that resists stretching, while lamins contribute little as they bend easily. Longer deformations strain nuclear lamins. (**C**) During migration, the nucleus protrudes into the pore resulting in large deformations that necessitates and activates lamin A resistance to maintain shape stability. Shape disruption can result in nuclear ruptures that lead to DNA damage [279]. Created with BioRender.com.

**Table 1 ijms-22-11289-t001:** Emerin binding partners. Emerin regulates a number of cellular pathways by acting directly or indirectly with key players in these pathways. Reported emerin binding partners are grouped according to their known or proposed functions.

Mechano- transduction	Nuclear Structure	Gene Regulation	Chromatin Tethering	References
Nesprin-1α	Lamin A/C	GCL	HDAC3	[59,65,76,157]
Nesprin-2β	Lamin B	Btf	BAF	[62,65]
SUN1	Actin	Lmo7	Lamin A/C	[53,63,125]
SUN2	Nuclear Myosin I	β-catenin	Lamin B	[139]
Lamin A/C	MAN1	YT521-B		[79,204]
SAMP1	αII-spectrin	Msx1		[65,180]
	β-dystroglycan	SIKE		[65,205,206]

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
