# Peer review of "The Role of Emerin in Cancer Progression and Metastasis"

_ijms, 2021, doi:10.3390/ijms222011289_

Round 1
Reviewer 1 Report
The review from Liddane and Holaska shows a comprehensive study of the role of Emerin in several aspects of the nuclear envelope role. It highlights the complex NE regulation and deregulation observed in cancer. It is a very interesting review that I enjoyed reading.
I have only few comments to make:
- line 101: ‘Emerin diffuses freely through the NPC’. Do the authors mean INM instead of NPC? Just to confirm.
- Line 637: ‘Micronuclei can form as a result of NE rupture’. The authors reference the paper of Vargas et al, where they describe for the first time the NE disruption during interphase and mention the presence of ‘micronuclei-like structures’ that are more a NE blebbing. Micronuclei are fully detached from the nucleus and are formed during mitosis as a result of lagging chromosome or the presence of a broken chromosome. NE disruption of the nucleus leads to massive DNA damage but do not release DNA out of the nucleus. A recent paper from Piel lab just came out and show that the exonuclease TREX1 can localize inside the nucleus after the transient NE disruption that can lead to massive DNA damage. (https://doi.org/10.1016/j.cell.2021.08.035).
- Cells treated with TGFb, allowing their transient conversion EMT are associated with NE deformability and NE protein modulation such as Lamins and Emerin (https://doi.org/10.1016/j.celrep.2016.11.022). The transient modulation of NE protein during the metastasis process is maybe worth mentioning.
Reviewer 2 Report
This is an interesting review focused on the role of mechanical properties of the nuclear envelope (NE) in cancer. The authors convey an idea that emerin and other NE components are important players in determining the metastatic potential of cancer cells. The review also contains detailed information on the role of protein complexes associated with the NE. Some parts of the review are written in not fully correct English. Below, I provide numerous (but not exhaustive) examples of errors or misprints in the text. I have only minor comments that the authors should consider prior to publication of this review:
- The authors reviewed functions of emerin based mainly on the data from mammals. However, they may wish to more widely employ an information concerning emerin from other organisms, such as Drosophila and nematode. Any case, the organism should be clearly indicated when authors describe properties of emerin and other NE proteins.
- 1: The legend does not completely describe what is depicted on the figure. In particular, there are no ER, ONM and INM abbreviations on the figure which are present in the legend, whereas there are many other factors depicted which are not deciphered in the legend.
- Line 69: “There are four lamin isoforms: lamin A, Lamin B1, Lamin 69 B2 and Lamin C.” It should be indicated that this statement is related only to mammals, since in Drosophila, for example, there are two lamin isoforms.
- Line 73: “The lamina networks in the nucleus.” Is it a heading? Or should it be deleted?
- Line 78: “although increased expression of lamin B1 can also increases nuclear rigidity” change to “although increased expression of lamin B1 can also increase nuclear rigidity”.
- Line 91: “These findings are particularly relevant to invading cancer cells, which must move through tissues with pore sizes smaller than the size of the nucleus…” Please, explain what does “pore sizes” mean in this context?
- Line 102: “[39, 40]and” change to “[39, 40] and”.
- Line 111: “NE enrichment [51]” change to “NE enrichment [51].”
- Line 113: “assembling nuclear envelopes” change to “assembling nuclear envelope”.
- Line 114: “BAF is a key NE localization factor both emerin and lamin A” change to “BAF is a key NE localization factor for both emerin and lamin A”.
- Line 123: “the interaction between the LEM domain and the IDR are necessary” change to “the interaction between the LEM domain and the IDR is necessary”.
- Line 135: “One set targeted residues was identical between emerin and LAP2β” change to “One set targeted residues that were identical between emerin and LAP2β”.
- Line 151: “Emerin Function” change to “Emerin Functions”.
- Line 159: “GCL is a transcription repressor binds and inactivates the DP3 subunit” change to “GCL is a transcription repressor which binds and inactivates the DP3 subunit”.
- Line 170: “via GCL- binding” change to “via GCL binding”.
- Lines 172-182: I recommend to rewrite this paragraph in a more straightforward manner to better convey the ideas of the authors.
- Lines 190-191: “Emerin inhibits Lmo7 activation by binding to Lmo7 to inhibiting Lmo7 binding to the promoters of the myogenic differentiation genes [83].” I do not understand this sentence.
- Line 195: “Lmo7 is predicted to play important roles muscle cell adaptation to mechanical stress.” change to “Lmo7 is predicted to play important role in muscle cell adaptation to mechanical stress.”
- Line 196: “Lmo7 also has been well documented role in cancer pathology.” change to “Lmo7 also has well documented role in cancer pathology.”
- Line 199: “and promoted migration of cells in vitro invasion assay” change to “and promoted migration of cells in in vitro invasion assay”.
- Line 226-227: “Mutations in BAF that cause a decrease in BAF protein expression due to protein instability causes Nestor-Guillermo progeria syndrome (NGPS)” change to “Mutations in BAF that lead to a decrease in BAF protein expression due to protein instability cause Nestor-Guillermo progeria syndrome (NGPS)”.
- Line 272: “Interesting,” change to “Interestingly,”.
- Line 275: “[131-134, 142] [108-111, 118, 119].” change to “[131-134, 142, 108-111, 118, 119].
- Line 296: “[126] [120, 127]” change to “[120, 126, 127]”.
- Line 311: “There is overwhelming evidence supporting the role in the mechanical support of the nucleus.” change to “There is overwhelming evidence supporting the role of nuclear lamina in the mechanical support of the nucleus.”
- Line 333: “Nesprins interacts at the C-terminus of INM SUN-domain proteins” change to “Nesprins interacts with the C-terminus of INM SUN-domain proteins”.
- Line 336: “The interactions between SUN-domain proteins and nesprins maintain the size of the lumen [13, 30] and aids in the positioning of the nucleus” change to “The interactions between SUN-domain proteins and nesprins maintain the size of the lumen [13, 30] and aid in the positioning of the nucleus”.
- Line 356: “This domain consisting of a transmembrane domain and a short, luminal domain, essential for anchoring nesprins to the NE [136] [23].” change to “This domain consisting of a transmembrane domain and a short, luminal domain, is essential for anchoring nesprins to the NE [136] [23].”
- Lines 377-380: “An additional mechanism lamins and emerin can affect mechanotransduction signaling was identified, revealing the actin polymerization-promoting activity of emerin at the NE can influence nuclear and cytoskeletal actin dynamics to modulate localization and activity of the mechanosensitive transcription factor, MKL1 [120].” change to “An additional mechanism by which lamins and emerin can affect mechanotransduction signaling was identified. The actin polymerization-promoting activity of emerin at the NE can influence nuclear and cytoskeletal actin dynamics to modulate localization and activity of the mechanosensitive transcription factor, MKL1 [120].”
- Lines 400-401: “The loss of emerin or 4.1R leads to a loss in INM retention of the other protein, suggesting the localization of either protein is mutually dependent”. I don’t understand what is written in this sentence.
- Line 406: “Many studies have shown emerin-null cells have less repressed chromatin.” Change to “Many studies have shown that emerin-null cells have less repressed chromatin.”
- Line 410: “Together, this data supports a role” change to “Together, these data support a role”.
- Lines 418-420: “Global mapping of chromatin interactions with B-type lamins or emerin showed approximately 40% of the Drosophila and human genomes contact the NE in large (0.1-10Mb [193]) discrete regions named lamin-associated domains (LADs) [193, 194].” LADs in Drosophila were first identified in the work of van Bemmel et al. (2010, doi: 10.1371/journal.pone.0015013), but not in the cited work of Pickersgill et al. (2006, ref. 194).
- Lines 422-423: “LAD organization is highly dynamic and can be altered in response to extracellular signaling and cell differentiation [200, 201].” There is no information about LADs dynamic in the work of Mehta et al. 2010 (reference 201), where positiones of whole chromosomes were assayed. Besides mammals (ref. 200), such type of information was presented in the work of Pindyurin et al. (2018, doi: 10.1186/s13072-018-0235-8), where LADs were identified and compared in various Drosophila tissues.
- Table 1: It is customary to provide references on the corresponding statements in the last column of the Table.
- Line 480: “or expressing mutant NE proteins such [207, 208]” change to “or expressing mutant NE proteins [207, 208]”.
- Line 497: “[218][219]” change to “[218, 219]”.
- Line 505: “It is expected changes in the nuclear architecture will alter” change to “It is expected that changes in the nuclear architecture will alter”.
- Line 506-507: “In cancer, increased nuclear deformability may allow metastatic cells to pass through narrow constructions more readily” change to “In cancer, increased nuclear deformability may allow metastatic cells to pass through narrow space between cells more readily”.
- Line 523: “was show” change to “was shown”.
- Lines 529-530: “A proteomic analysis of soft and stiff tissues revealed A-type lamins increases with tissue stiffness” change to “A proteomic analysis of soft and stiff tissues revealed that A-type lamins increases with tissue stiffness”.
- Line 538: “This supports a role of A-type lamins as a function of the mechanical environment.” I do not understand this sentence.
- Line 544: “both the structural and the levels of A-type lamins” change to “both the structural organization and the levels of A-type lamins”.
- Line 554: “Altered lamin A/C levels would be expected in enhanced stiffness” change to “Altered lamin A/C levels would be expected upon enhanced stiffness”.
- Line 559: “ These finding support the hypothesis tissue rigidity promotes invasion” change to “These findings support the hypothesis that tissue rigidity promotes invasion”.
- Line 564: “One study found mouse mammary tumors had increased metastasis with more compliant tumors …” I do not understand this sentence.
- Line 599: “They found in the absence of proteolysis …” change to “They found that in the absence of proteolysis …”
- Line 671: “Longer deformations strains nuclear lamins.” change to “Longer deformations strain nuclear lamins.”
- Line 690: “Both explanations results in altered nuclear mechanics” change to “Both explanations result in altered nuclear mechanics”.
- Line 705: “Emerging evidence shows …” change to “Emerging evidence show …”.
- Lines 742-1231: Reference list is not formatted according to the IJMS requirements. In particular, “et al.” should be written if more than ten authors are in the reference. Title of articles should not be italicized. And so on.
